# Synergistic Effects of Melittin and Plasma Treatment: A Promising Approach for Cancer Therapy

**DOI:** 10.3390/cancers11081109

**Published:** 2019-08-03

**Authors:** Priyanka Shaw, Naresh Kumar, Dietmar Hammerschmid, Angela Privat-Maldonado, Sylvia Dewilde, Annemie Bogaerts

**Affiliations:** 1Research Group PLASMANT, Department of Chemistry, University of Antwerp, BE-2610 Wilrijk-Antwerp, Belgium; 2Laboratory of Protein Science, Proteomics & Epigenetic Signaling, Department of Biomedical Sciences, University of Antwerp, BE-2610 Wilrijk-Antwerp, Belgium

**Keywords:** melittin, reactive oxygen and nitrogen species, oxidation, molecular dynamics, permeation free energy, cold atmospheric plasma

## Abstract

Melittin (MEL), a small peptide component of bee venom, has been reported to exhibit anti-cancer effects in vitro and in vivo. However, its clinical applicability is disputed because of its non-specific cytotoxicity and haemolytic activity in high treatment doses. Plasma-treated phosphate buffered saline solution (PT-PBS), a solution rich in reactive oxygen and nitrogen species (RONS) can disrupt the cell membrane integrity and induce cancer cell death through oxidative stress-mediated pathways. Thus, PT-PBS could be used in combination with MEL to facilitate its access into cancer cells and to reduce the required therapeutic dose. The aim of our study is to determine the reduction of the effective dose of MEL required to eliminate cancer cells by its combination with PT-PBS. For this purpose, we have optimised the MEL threshold concentration and tested the combined treatment of MEL and PT-PBS on A375 melanoma and MCF7 breast cancer cells, using in vitro, in ovo and in silico approaches. We investigated the cytotoxic effect of MEL and PT-PBS alone and in combination to reveal their synergistic cytological effects. To support the in vitro and in ovo experiments, we showed by computer simulations that plasma-induced oxidation of the phospholipid bilayer leads to a decrease of the free energy barrier for translocation of MEL in comparison with the non-oxidized bilayer, which also suggests a synergistic effect of MEL with plasma induced oxidation. Overall, our findings suggest that MEL in combination with PT-PBS can be a promising combinational therapy to circumvent the non-specific toxicity of MEL, which may help for clinical applicability in the future.

## 1. Introduction

Melittin (MEL) is a water-soluble cationic amphipathic 26 amino acid α-helical peptide obtained from the honeybee (*Apis mellifera*) venom [1]. It is a very nonspecific cytolytic peptide that rapidly associates with phospholipid cell membranes. It moves in a lateral direction in the membrane, yielding oligomerization, thereby leading to structural defects (e.g., pores) in the cell membrane [2]. In addition, when reaching the intracellular environment, it can act in a similar way on the membrane of internal organelles, inducing biochemical changes that cause cell death [3,4]. Therefore, several studies have demonstrated that MEL has inhibitory effects on the proliferation of various cancer cells in vitro via the induction of apoptosis, necrosis and cell lysis [5]. MEL can target a range of cancer cells, including those in leukaemia, lung, renal, liver, bladder and prostate cancer, via activation of a caspase-dependent pathway [6,7,8,9]. However, despite the convincing efficacy data against various cancers, its clinical applicability is precluded due to the non-specific toxicity shown at high doses. Specifically, in vivo experiments have demonstrated that MEL can induce cytolysis, aggregation of membrane proteins, haemolytic activity, spontaneous pain, increased blood flow (neurogenic inflammation) and the appearance of regions of hyperalgesia around the site of injection [10,11,12,13]. These toxic aspects of MEL are indeed considered as a limiting factor for its use in cancer therapy [14]. Nevertheless, to reduce the nonspecific toxicity, several combinations of MEL with chemotherapeutic drugs and nanotechnology have been reported [1,15,16,17]. However, these combinations still remain challenging [18]. Additionally, Orsolic and Alonezi et al. explored the dose dependent growth-inhibiting impact of MEL in conjunction with cytotoxic drugs such as cisplatin and bleomycin on melanoma, HeLa and V79 cells in vitro [19,20]. Further, Alizadehnohi et al. reported that MEL enhanced the cytotoxic impact of cisplatin in human ovarian cancer cells [21]. However, the combined use of MEL and cisplatin to treat cancer cells still remained a challenge due to the side effects and off-target toxicity [18,22]. It has been suggested that the combination of MEL with nanoparticles could increase the safe delivery of significant amounts of MEL intravenously to target and kill tumours, while reducing the haemolytic activity of MEL [1,15]. However, the role of nanotechnology in delivering MEL is still at its early development stage because of drawbacks during the preparation for nano delivery systems such as aggregation, morphological changes, peptide stability, etc. Moreover, these systems are expensive to implement for cancer therapy [23]. Thus, further studies aiming to reduce the therapeutic dose of MEL and its associated unspecific cytolytic activity are needed. In this context, we propose the combination of MEL with a solution treated with cold atmospheric plasma (CAP), a novel therapy that could help to overcome the current limitations of MEL.

CAP is a partially ionized gas, which contains a mixture of highly reactive chemical species, also called reactive oxygen and nitrogen species (RONS), such as ^•^OH, O, O_2_^•−^, ^1^O_2_, O_3_, NO^•^, NO_2_^•^, NO_2_^−^, NO_3_^−^ and H_2_O_2_ [24,25]. CAP can react with an organic surface without inflicting any thermal or electrical damage, and recently it has been shown that it can selectively target cancer cells with minimal effects on normal cells [26]. Thus, CAP is now being investigated and used in various medical applications, ranging from sterilization, antifungal treatment, tooth bleaching and chronic wound healing, to cancer therapy [27,28,29]. Usually it is believed that CAP-produced RONS can enhance the fluidity of the cell membrane through lipid peroxidation, which eventually affects the intracellular biochemical signalling pathways [30,31]. In addition, it has been shown that CAP facilitates the uptake of nanoparticles and enhances their therapeutic action in cells [32]. For instance, it was shown that gold nanoparticles were endocytosed at an accelerated rate in the U87 cell membrane due to the RONS generated by CAP [33].

In the present study, we have used in vitro, in ovo and in silico approaches to study the ability of CAP-treated phosphate buffered saline solution (PT-PBS) to reduce the nonspecific toxicity of MEL and to induce cell death in A375 melanoma and MCF7 breast cancer cells. To investigate the cytotoxic effect, various ratios of PT-PBS and MEL alone and in combination were applied to both cell lines. After dose optimisation of MEL and PT-PBS alone and in combination, we evaluated cell death and lipid peroxidation.

Additionally, to obtain a better insight into the level of synergy of MEL and PT-PBS, we calculated the free energy barrier for the translocation of MEL across native and (plasma-)oxidised phospholipid bilayers (PLBs) through molecular dynamics (MD) simulations. In the literature, MD simulations have reported the interaction of MEL with PLB [34], the peptide orientation [35], the deformation of lipids [36], and the change in secondary structure [37]. MEL has been shown to maintain a stable helical structure on the PLB surface [38]. However, only a few articles have reported the simulation of the free energy of MEL in a native PLB [39,40]. Irudayam et al. [39] studied the free energy barrier for MEL reorientation from a membrane-bound state (i.e., with the MEL helix parallel to the surface of the PLB) to a transmembrane state (or T-state, i.e., with the MEL helix perpendicular to the bilayer surface in the lipid region of the PLB). They found that a higher MEL concentration leads to a decrease of the free energy barrier. Furthermore, Irudayam and Berkowitz [40] calculated the free energy of MEL adsorption on the PLB surface. The authors indicated that MEL expresses a strong affinity to the bilayer surface. Note that the above studies only applied to a native PLB, and not to oxidized PLBs, which would be the result of plasma oxidation. Indeed, the translocation of MEL through the PLB, and the free energy barrier of MEL in an oxidized PLB was not yet investigated. Therefore, in the present study we analysed the free energy barrier for translocation of MEL in both native and (plasma-) oxidized membrane. Overall, our experimental and computational approaches reveal that this combinational therapy opens unique opportunities for future cancer treatment.

## 2. Materials and Methods

### 2.1. Reagents and Cell Lines

The following reagents and kits were used in this study: MTT (3-[4,5-dimethylthiazol-2yl]-2,5-diphenyltetrazolium bromide) (Sigma-Aldrich, Darmstadt, Germany), DMSO (dimethyl sulphoxide) (Sigma-Aldrich), Annexin V-FITC apoptosis detection kit (BD Biosciences, Allschwil, Switzerland), Image-iT (Thermo Fisher Scientific, Merelbeke, Belgium), ProLong™ Gold Antifade Mountant with DAPI (Thermo Fisher Scientific), MDA Assay Kit (CELL BIOLABS, INC, San Diego, CA, USA). Melittin 85% (HPLC) was purchased from Sigma-Aldrich. The human breast adenocarcinoma (MCF7) and human malignant melanoma (A375) were obtained from the American Type Culture Collection (ATCC, Manassas, VA, USA).

### 2.2. Plasma Device and Sample Preparation

MEL 85% (HPLC) was purified from bee venom and reconstituted in sterile PBS to form a stock solution of 1 mg/mL before storage at −20 °C until required. The concentration of MEL mentioned in the following text (0–10 µL/mL) means that 0–10 µL of the original stock solution were added to cells contained in 1 mL culture media. In this study, the abbreviations MEL-10, MEL-5, MEL-2.5, MEL-1.2, and MEL-0.6 are used for the MEL concentrations of 10 µg/mL, 5 µg/mL, 2.5 µg/mL, 1.2 µg/mL, and 0.6 µg/mL, respectively. To make a plasma-treated PBS (PT-PBS) solution, the kINPen^®^ IND plasma jet (INP Greifswald/neoplas tools GmbH, Greifswald, Germany) was used, as shown in Figure 1.

Detailed information about the design and operation of the kINPen^®^ IND plasma device was reported previously [41,42]. Briefly, the kINPen^®^ IND plasma device houses two electrodes: a pin electrode (1 mm diameter) in the centre that is separated by a dielectric capillary (1.6 mm inner diameter) from a grounded ring electrode. The plasma is generated by high frequency sinusoidal voltage of around 2–6 kVpp to the central electrode, with a frequency between 1.0 and 1.1 MHz and a maximum power of 3.5 W. To control the temperature, the device operates in switch off/on mode with a frequency of 2.5 kHz. The plasma is ignited inside the capillary and due to the gas flow it creates a plasma effluent towards the open side of the device, with a length of 9–12 mm and ca. 1 mm diameter [41,42]. For the plasma treatments, the plasma source was fed by argon gas for 9 min to treat 2 mL PBS (pH 7.3) in a 12-well plate. During the plasma treatment we kept a 10 mm distance between the nozzle of the plasma jet device and the liquid surface, as shown in Figure 1 [24,42]. The RONS present in plasma-treated PBS have been previously described [24], being H_2_O_2_ and NO_2_^−^ as two of the main RONS responsible of the biological effects of PT-PBS.

### 2.3. Analysis of Cell Cytotoxicity

The cytotoxicity of PT-PBS and MEL on breast adenocarcinoma (MCF7) and human malignant melanoma (A375) was evaluated. Both cell lines were cultured in Dulbecco’s Modified Eagle Medium (DMEM; Life Technologies, Merelbeke, Belgium) supplemented with 10% (*v*/*v*) fetal bovine serum (Life Technologies) and 1% (*v*/*v*) penicillin and streptomycin (Life Technologies). The cell cultures were maintained at a temperature of 37 °C in a humidified incubator containing 5% (*v*/*v*) CO_2_. The cells were grown in 75 cm^2^ tissue culture flasks until they reached confluence and then they were sub-cultured for the experiments. Seeded into 24-well plates to grow in complete media were 2 × 10^5^ cells/wells. After 24 h for the single treatment with PT-PBS and MEL, the medium was removed and immediately replaced with media supplemented with different percentages of PT-PBS, i.e., 20%, 10%, 5%, 2.5% and 1.2%, and different concentrations of MEL, i.e., MEL-10, MEL-5, MEL-2.5, MEL-1.2, and MEL-0.6 (as described above). Subsequently, to obtain the synergetic effect of both treatments, the optimized doses of the single treatments (PT-PBS and MEL) were combined. A total volume of 1000 µL per well was maintained after addition of the respective percentages of PT-PBS and MEL. In this study, the abbreviations PT-PBS 20, PT-PBS 10, PT-PBS 5, PT-PBS 2.5 and PT-PBS 1.2 were used for the percentages of 20%, 10%, 5%, 2.5% and 1.2% of PT-PBS (formed after 9 min plasma treatment), respectively. The cells were placed in an incubator and their viability was monitored for 24 h post PT-PBS and MEL treatment. To check the viability, the MTT assay was performed after 24 h. In this assay, 50 μL MTT (3-[4,5-dimethylthiazol-2yl]-2,5-diphenyltetrazolium bromide) solution (5 mg/mL in PBS) was added to each well. After 3 h incubation, the purple formazan precipitates in each well were released in the presence of 1000 µL DMSO (dimethyl sulphoxide). The absorbance was measured using a microplate reader (BIO-RAD iMark Microplate reader, Temse, Brussel, Belgium) at 540 nm and the cell viability was assessed as the absorbance ratio between the treated and control sample, which was directly proportional to the number of metabolically active cells. To detect cell death, we used the Annexin V-FITC apoptosis detection kit. The treated/untreated cells were washed with 1 mL of cold 1× binding buffer after 3 h incubation and were subsequently trypsinized. Annexin V-FITC (0.5 mg/mL) was added to each sample. After incubation for 15 min at room temperature, the cells were again washed with PBS, stained with 0.3 mg/mL of PI (Propidium Iodide) and analysed using flow cytometry (Attune NxT Flow Cytometer, Brussels, Belgium).

### 2.4. Estimation of Lipid Peroxidation

After standardisation of the optimal dosages of PT-PBS, MEL alone and in combination through the MTT and flow cytometry analysis, we investigated the peroxidation of membrane lipids by fluorescence microscopy. We seeded 10^5^ cells on cover glass. After 24 h incubation, the cells were exposed to PT-PBS, MEL alone or to the combination of PT-PBS 10 and MEL-1.5. After treatment for 3h, the cells were incubated with 10 µM Image-iT (lipid peroxidation) molecular probe for 30 min and fixed in 4% paraformaldehyde (in PBS) for 20 min and then permeabilized in cytoskeleton buffer (pH 6.8, 50 mM NaCl, 150 Mm sucrose, 3 mM MgCl_2_, 50 mM Trizma-base, 0.5% Triton X-100). After permeabilization, the cells were washed thrice with PBS and subsequently Gold Antifade DAPI was used to mount the cells. The cells were imaged with Zeiss AxioImager Z1 microscope (Carl Zeiss, Göttingen, Germany) equipped with an AxioCam MR ver.3.0 using a 40× objective, using filters for green fluorescent (GFP), red (Texas Red) and blue fluorescent (DAPI) channels. Next, flow cytometry was used to quantify the fluorescence intensity, after 3 h incubation with PT-PBS, MEL alone and in combination, at 37 °C in a 5% CO_2_ atmosphere. The treated/untreated cells were incubated with 10 µM Image-iT molecular probe for 30 min, and then harvested by trypsinization, washed twice with PBS, and finally the cells were resuspended in PBS to detect the fluorescence intensity by flow cytometry. To quantify the fluorescence intensity, we used Texas Red^®^ (590 nm) and FITC (510 nm) emission filters, and we calculated the ratio of intensity in the Texas Red^®^ channel to the intensity in the FITC channel. Moreover, for all treatments we also measured the malondialdehyde (MDA) concentration by the MDA assay kit, following the standard protocol according to the manufacturer’s instructions. This method is based on the reaction of free MDA (present in the sample) with thiobarbituric acid (TBA) to generate an MDA-TBA adduct and its quantification is generally used as marker for lipid peroxidation.

### 2.5. Chicken Chorioallantoic Membrane Assay (CAM Assay)

Four-day old fertilized chicken eggs were incubated in a horizontal position for 1 day at 37.7 °C and 65% humidity in an egg incubator with automatic turning function (Ova-Easy 100, Brinsea, Veenendaal, The Netherlands). On day 5, the upper pole was disinfected and pierced with a 20G sterile needle (BD) and sealed with medical tape (Leukosilk S, Covamed Farma BVBA, Marke, Belgium). The eggs were incubated in vertical position (turning function off) to promote the relocation of the air cell. On day 7, the egg shell was cut to expose the chicken chorioallantoic membrane (CAM). A 1 × 1 mm filter paper soaked in diethyl ether (Fisher Scientific, Merelbeke, Belgium) was briefly applied on a vascularized region of the CAM and a sterile silicone ring (ID = 5 mm, OD = 6 mm) was placed. A pellet of A375 cells (2 × 10^6^ cells per egg) was mixed with 15 µL growth reduced factor Matrigel (8.6 mg/mL, Corning, Amsterdam, The Netherlands) and loaded into the ring. The eggs were sealed with Tegaderm (3D) and placed back in the incubator for 4 days. On day 11, the Tegaderm was cut and a sterile plastic ring (ID = 7 mm, OD = 8.5 mm) was placed around the tumour. One hundred micro litres of untreated PBS, MEL-1.5 µg in 100 µL PBS, 10% of PT-PBS and its combination (MEL + PT-PBS) were loaded into the ring. The eggs were sealed with Tegaderm and incubated until the end of the experiment. The cytotoxic effect of the treatments was assessed on day 14 when tumours were excised and weighed in a precision balance (Mettler Toledo, Fisher, Merelbeke, Belgium). All steps outside the incubator were carried out using a heat block (set at 37.7 °C) with a custom-made egg-shaped aluminium adapter.

### 2.6. Immunohistochemical Analysis for Ki-67

After weighing, the tumours were fixed with 4% paraformaldehyde for 14 h at 37 °C prior to paraffin embedding. Sections of 5 µm were cut, deparaffinized, rehydrated and stained with 1:1 haematoxylin and 0.5% eosin (HE) solution for histological analysis. For Ki-67 staining, antigen retrieval was performed with citrate buffer (10 mM, pH 6), at 96 °C for 20 min. Sections were permeabilised in 0.1% Tween-20 and blocked with 3% H_2_O_2_ in PBS (10 min, RT) and 2% BSA (30 min, RT). The primary antibody incubation was 40 min at RT (1/75 dilution; Mouse Anti-Human Ki-67 Antigen, Clone MIB-1, Agilent, Santa Clara, CA, USA), followed by incubation with the secondary antibody (30 min at RT; Envision Flex HRP, Agilent, Diegem, Belgium). Diaminobenzidine was used to visualize positive staining and haematoxylin to counterstain. Sections were imaged with a Zeiss AxioImager Z1 microscope (Carl Zeiss, Göttingen, Germany) equipped with an AxioCam MR ver.3.0. 

### 2.7. Mass Spectrometry Analysis

To detect the conformational change in MEL by native mass spectrometry (MS), 1 mg of MEL was dissolved either in 1 mL untreated PBS (control) or in 1 mL PT-PBS (i.e., 100% of PT-PBS). Each sample (50 µL) was buffer exchanged to a MS compatible 100 mM ammonium acetate solution using a Micro Bio-Spin 6 column (Bio-Rad, Hercules, CA, US). MS experiments were conducted on a Synapt G2 HDMS (Waters, Manchester, UK) instrument. For this purpose, 3 µL of sample was loaded into an in-house produced gold-coated borosilicate capillary and mounted onto the instrument. The sample was ionized by nano-electrospray ionization (nESI) and the generated ions were drawn into the vacuum of the instrument. The crucial parameter settings were 1.2 kV capillary voltage, 10 V sampling cone, 1 V extractor cone, 10 V and 2 V collision energy in the trap and transfer cell, respectively. Pressures throughout the instrument were set to: 2.75 mbar backing pressure, 4.55 × 10^−3^ mbar in the source region, and 2.5 × 10^−2^ mbar in the trap and transfer collision cells.

### 2.8. Statistical Analysis

Data was analysed using the Students’ *t*-test comparison analysis. The data was considered significantly different when * = *p* ≤ 0.05, ** = *p* ≤ 0.01, *** = *p* ≤ 0.001. All values represent experiments done in triplicates. Data shown as mean ± standard deviation (SD). Prism (Graphpad Software Inc., San Diego, CA, USA) and Excel Software (Microsoft Inc., Redmond, WA, USA) was used to compare the groups.

### 2.9. Computational Analysis

#### 2.9.1. Simulation Setup

To support the experiments, we performed MD simulations to study MEL translocation across native and oxidized PLBs. We chose the PLB as the model system for the cellular membrane, as it provides the structural framework for the cell membrane. The PLB considered in this study consists of palmitoyl-oleoyl-phosphatidylcholine (POPC) lipids (see Figure 2).

To study the effect of plasma-induced oxidation of the phospholipids on the translocation of MEL through the PLB, we assumed aldehyde oxidation products (POPC-ALD, see Figure 2b), which are found to be one of the key oxidation products [43]. The simulations were carried out using the GROMACS package (version 5.1) [44], applying the GROMOS 54A7 force field [45]. The force field parameters of the aldehyde product of the oxidized POPC (POPC-ALD) were obtained from [46]. To generate the initial configurations of the intact (or native) and oxidized POPC systems, we applied the Packmol package [47]. Each system consists of 20,000 water molecules together with 128 phospholipids organized in two layers (i.e., 64 lipids with corresponding water layer at the top, and 64 at the bottom, see Figure 2a). MEL has an α-helical configuration and was placed on the upper side of the PLB, i.e., in the water phase at about 1.5 nm above the head group region of the bilayer (see Figure 2a, and below for more details). To evaluate the effect of plasma-induced oxidation of the native POPC, we replaced randomly 64 POPC molecules with POPC–ALD (i.e., 32 at the top and 32 at the bottom), corresponding to 50% oxidation. Thus, we studied two model systems, i.e., native (0% oxidation) and 50% aldehyde-oxidized PLBs. We assumed 50% oxidation, which is enough to clearly investigate the effect of translocation of MEL across the bilayer, but low enough to avoid pore formation within the simulation [48]. In order to obtain the average free energy profile (FEP) of MEL translocation across each system (see section below), we created three model systems for both native and 50% oxidized PLB. In each system we changed the position of MEL in the xy-plane, keeping the distance between MEL and the centre-of-mass (COM) of the bilayer constant in the z-direction (reaction coordinate). Moreover, to neutralize the system, we added 6 Cl^−^ ions, because of the +6 charge of MEL. After construction of the hydrated membranes with equilibrated MEL at the top, all the systems were energy-minimized using the steepest descent algorithm. Further, all structures (i.e., three native and three oxidized PLBs) were equilibrated for 200 ns (for the native case) and 300 ns (for the oxidized case) in the NPT ensemble (i.e., at constant number of particles, pressure and temperature), at 310 K and 1 bar, employing the semi-isotropic Parrinello–Rahman barostat [49] with a compressibility and coupling constant of 4.5 × 10^−5^ bar^−1^ and 1 ps, respectively, as well as Nose-Hoover thermostat [50] with a coupling constant of 0.2 ps. For the non-bonded interactions, a 1.2 nm cut-off was applied. Periodic boundary conditions were applied to all systems in all Cartesian directions. The long-range electrostatic interactions were described by the particle mesh Ewald (PME) method [51], using a 1.2 nm cut-off for the real-space interactions and 0.15 nm spaced-grid for the reciprocal-space interactions. The SPC/E (extended simple point charge) model was used to represent the water molecules surrounding the membrane and MEL. In all simulations, we used a time step of 2 fs. 

#### 2.9.2. Umbrella Sampling

In order to determine the FEPs of MEL translocation through the native and 50% oxidized PLBs, we applied umbrella sampling (US) simulations. To avoid disturbances in the hydrophobic part of the bilayer, we kept MEL perpendicular to the surface (as shown in Figure 2a), also called the transmembrane state (T state), which is the stable state of MEL when it begins to diffuse inside the membrane [52]. Furthermore, to avoid the formation of the U-shaped conformation of MEL, we pulled the COM of the first three residues of the N-terminus of MEL towards the bilayer (see arrow indicated in Figure 2a).

For each FEP, we extracted 95 windows along the z-axis, which were separated by 0.1 nm. These windows were obtained by pulling the COM of the first three residues of the N-terminus of MEL in the z-direction (as mentioned above), applying a harmonic bias between MEL and the COM of the PLB, with a force constant of 2000 kJ·mol^−1^ nm^−2^ and a very slow pulling rate of 0.001 nm·ps^−1^. Each US simulation lasted for 200 ns, and the last 50 ns were used to collect the US histograms and to calculate the FEPs. A periodic version of the weighted histogram analysis method (WHAM) [53] implemented in GROMACS, was applied to construct the FEPs. The final energy profiles were obtained by averaging over three FEPs for each system, which differ from one another based on their starting structure, to allow for some statistical variations. 

## 3. Results and Discussion

### 3.1. Effects of PT-PBS and MEL on Cell Viability and Dose Optimization

To verify the effect of MEL, PT-PBS alone and its synergy on the growth of melanoma and breast cancer cells, we analysed the cell viability using the MTT assay. Treatments by MEL and PT-PBS alone exerted a concentration-dependent cytotoxic effect on both cell lines (Figure 3a,b). Up to 2.5 µg/mL of MEL shows significant decrease in viability in both cell lines. In addition, PT-PBS 20 and PT-PBS 10 also show a significant inhibitory effect on both cell lines after incubation for 24 h. For the controls, 20% and 10% non-treated PBS in the culture media were used for the MTT assay for the A375 and MCF7 cells, respectively, and 10% non-treated PBS in culture media for the cell death analysis and lipid peroxidation analysis in both cell lines.

The half maximal inhibitory concentrations (IC_50_) of MEL and PT-PBS are shown in Figure 3c. The IC_50_ values of MEL in A375 and MCF7 cells were 2.5 and 2.8 µg/mL, respectively, while they were 16.4% and 16.5%, respectively, for PT-PBS. Based on the IC_50_ values, we performed the combined treatment of MEL and PT-PBS, to investigate the synergy between both. For this purpose, we fixed the concentration of PT-PBS to a value lower than its IC_50_ values (i.e., we fixed it at 10%) and we varied the concentrations of MEL in a range lower than its obtained IC_50_ values (i.e., 2, 1.5, 1 and 0.5 µg/mL).

As shown in Figure 4a,b, PT-PBS 10 alone yielded 65% viability for both A375 and MCF7 cells, while MEL-2, MEL-1.5, MEL-1 and MEL-0.5 alone resulted in ca. 50%, 60%, 77% and 90% cell viability for the A375 cells, and ca. 55%, 70%, 80% and 95% for the MCF7 cells, respectively. However, the combination of both showed a significant (*p* < 0.001) decrease, to ca. 8%, 15%, 32% and 60% for the different MEL concentrations in the A375 cells, and to ca. 10%, 17%, 34% and 64% for the different concentrations in the MCF7 cells. 

To determine the synergistic cytotoxic effect of the combination of PT-PBS and MEL, the combination index (CI) value was calculated based on [54]. It is generally accepted that CI values < 0.1 indicate very strong synergism, CI = 0.1–0.3 strong synergism, CI = 0.3–0.7 synergism, CI = 0.7–0.9 slight synergism and CI = 0.9–1.1 nearly additive, while CI = 1.1–1.45 refers to slight to moderate antagonism [54,55]. As illustrated in Figure 4c, the CI analysis on A375 cells shows a synergistic cytotoxic activity for the combination of PT-PBS 10 with the following concentrations of MEL-2 (CI = 0.384), MEL-1.5 (CI = 0.412), MEL-1 (CI = 0.812), and MEL-0.5 (CI = 1.012). For the MCF7 cells, the CI values are very similar, i.e., MEL-2 (CI = 0.372), MEL-1.5 (CI = 0.426), MEL-1 (CI = 0.846), and MEL-0.5 (CI = 1.02). The combination of PT-PBS 10 with MEL-2 or MEL-1.5 clearly yields synergism, indicating that this combination can reduce the toxicity dose of MEL, and thus avoid the side effects related to higher doses of MEL. However, literature shows that MEL concentrations of up to 2 μg/mL do not significantly inhibit cell viability in melanoma and lung cancer [22,27,56]. Thus, we performed our further experiments with a low dose, i.e., MEL-1.5, in combination with PT-PBS 10, as the optimal combination for both cells. 

### 3.2. Influence of PT-PBS and MEL on Cell Death

To further demonstrate the synergism, and for clearer observation of cancer cell death after treatment with PT-PBS and MEL alone and in combination, we performed flow cytometry analysis. We also analysed the degree of apoptosis/necrosis after treatment of both cell types. FITC Annexin V and PI negative cells are considered as viable; if the cells are FITC Annexin V positive and PI negative that cells are considered as early apoptotic; however, if cells are both FITC Annexin V and PI positive that cells are considered as late apoptotic or already dead (by necrosis). Hence, this assay does not differentiate between apoptotic and necrotic cell death as both populations are positive for both FITC Annexin V and PI. Therefore, in the current experiments, Annexin V and PI positive staining represent late apoptotic combined with necrotic cell death [57].

As shown in Figure 5a,c for A375 cells, PT-PBS 10 and MEL-1.5 alone induced 38% and 35% late apoptosis/necrosis (Annexin V positive, PI positive), respectively, while the combined exposure induced 96% late apoptosis/necrosis (*p* ≤ 0.01). In a similar way, PT-PBS 10 and MEL-1.5 alone induced 37% and 30% late apoptosis/necrosis in MCF7 cells, respectively, while their combination induced 92% late apoptosis/necrosis (*p* ≤ 0.01, Figure 5b,d). These results indicate that the combined action of PT-PBS 10 and MEL-1.5 on both cancer cell lines is more than the sum of their individual effects, which is in agreement with the CI values shown above (Figure 4c). Altogether, these findings support the synergistic cytotoxic action of PT-PBS 10 and MEL-1.5 in cancer cells.

### 3.3. Effect of PT-PBS and MEL on Lipid Peroxidation

Lipid peroxidation generally refers to the oxidative degradation of cellular lipids by reactive oxygen species. Peroxidation of unsaturated lipids affects cell membrane properties [58] and signal transduction pathways [59]. Thus, to evaluate the change in membrane integrity upon treatment with PT-PBS and MEL, we estimated the lipid peroxidation with the MDA assay and fluorescent probe.

As shown in Figure 6a, the A375 cells incubated with PT-PBS 10 and MEL-1.5 alone yielded ca. 5.2 µM and 4 µM peroxidation product (malondialdehyde; MDA), respectively, while in combination they exhibited a significant formation of MDA (ca. 15 µM, *p* ≤ 0.01). Likewise, for the MCF7 cells, PT-PBS 10 and MEL-1.5 alone produced ca. 4.5 µM and ca. 4 µM MDA respectively, while their combined exposure shows a significantly higher MDA production (ca. 10.5 µM, *p* ≤ 0.05). In addition, lipid peroxidation following PT-PBS 10, MEL-1.5 and combined exposure was quantified in both cancer cells, using the ratio of green and red fluorescence intensities, as shown in Figure 6b. The ratios increased significantly in both cancer cells when we applied the combined treatment (*p* ≤ 0.05).

Figure 6c,d show the fluorescence images of the A375 and MCF7 cells stained with lipid peroxide detection reagents, after addition of PT-PBS 10, MEL-1.5 alone and in combination. Compared with the separate treatments, the combined exposure shows most of the signal is in the green channel, which indicates lipid peroxidation. We also detected lipid peroxidation products around the membrane, indicating a change in the physical properties of the cellular membranes. This can cause covalent modification of proteins and nucleic acids, which might eventually be critical mediators of oxidative stress-mediated cell death [59]. Thus, the combined treatment with PT-PBS 10 and MEL-1.5 induces lipid peroxidation in both cancer cell lines, which correlates with the cytotoxic effect of the combined treatment observed above.

### 3.4. Effect of PT-PBS and MEL on Malignant Melanoma Cancer Tumors of the TUM-CAM Model

Malignant solid tumours were analysed macroscopically by using HE and Ki-67 staining. The stained tumour tissue sections displayed changes in morphology upon all treatments. Interestingly, the combined treatment induced more cellular shrinking and presence of pyknotic dark small nuclei as a result of chromatin condensation, compared to the PT-PBS 10 and MEL-1.5 treatments alone (Figure 7a). In contrast, untreated (control) cells presented a more prominent malignant phenotype with mitotic activity (HE staining, Figure 7a). To assess the proliferative state of cells in the treated tumours, tissue sections were stained for the proliferation marker Ki-67. The untreated tumours presented the highest levels of Ki-67 positive cells. We observed that tumours exposed to the combined treatment presented the lowest number of Ki-67 positive cells, followed by tumours treated with PT-PBS 10 and MEL-1.5 alone (Figure 7b). These results suggest that the combined treatment of MEL+PT-PBS has a detrimental effect on cell proliferation, as it reduced the number of Ki-67 positive cells more efficiently than the sum of the individual treatments.

In agreement with the histological analysis of tumour specimens, the tumour weight also demonstrated a reduction of tumour size upon treatment (Figure 7c,d). The combined treatment significantly reduced the tumour weight by approx. 76%, whereas the tumours treated with PT-PBS 10 and MEL-1.5 showed a reduction in weight of approx. 30% and 35%, respectively (Figure 7c). Hence, these results indicate that the combined treatment induced a synergistic reduction of cell proliferation, as its effect on tumour weight (76% reduction) was approximately 11% higher than the sum of both treatments (reductions of 30% + 35%). These findings are in agreement with the in vitro results shown above. 

### 3.5. Plasma Oxidation of MEL: Mass Spectrometry Analysis

To better understand the synergistic effect of PT-PBS and MEL, we investigated the effect of plasma treatment on the oxidation level of MEL. For this purpose, we dissolved MEL either in untreated (control) or in PT-PBS buffer and we incubated the solution for 1, 60, or 120 min. Figure 8 shows the native MS measurements of the control (a) and the plasma-treated (b) samples. 

The indices refer to the different time points of incubation (0: 1 min, 1: 60 min, 2: 120 min). As illustrated, the comparison between control and plasma treatment reveals no differences in the oxidation level of MEL. The observed oxidations (700–750 *m*/*z*) may originate from the electrospray ionization [60], as indeed reported for MEL [61], but at much harsher operating conditions than applied in this experiment. Moreover, the measurements do not show an increase in the number of oxidations over time. Thus, no differences were observed in the spectra acquired after one minute and after two hours. Independent of whether the oxidations are already present in solution or are induced by the electrospray ion source, in any case the plasma treatment applied in the form of solubilizing MEL in plasma-treated PBS buffer does not have any influence on these observed oxidations. Hence, we can conclude that MEL is not oxidized by plasma, and that the synergistic effects of MEL and PT-PBS must be attributed to other effects. One possibility is the enhanced translocation of MEL through the cell membrane upon plasma-induced oxidation of the phospholipids. This will be discussed in the next section.

### 3.6. MEL Translocation across Native and Oxidized Phospholipid Membrane Revealed through MD Simulations

To gain further insight into the experimentally observed synergetic effect between MEL and the RONS present in PT-PBS and to investigate the mechanisms of MEL access to cancer cells through oxidized membranes, we performed US MD simulations. This method allows us to elucidate the MEL translocation across the native and oxidized PLBs, based on the FEPs plotted in Figure 9.

Based on the polarity of the lipids, the hydrophilic part refers to the upper and lower lipid head groups (light blue colour in Figure 9a), while the hydrophobic part refers to the lipid tails of the PLB (grey colour in Figure 9b–e). In the native case, when MEL enters from the water phase to the hydrophilic head group of the upper leaflet (see Figure 9b,c), it shows a high affinity with the (charged) head groups of the PLB, due to strong Coulomb interaction [62]. As a result, the potential of mean force for insertion of MEL shows a drop in the free energy barrier, making insertion of MEL favourable (see first minimum near *z* = 1.5–2 nm in Figure 9a, indicated with the arrow). Subsequently, the N-terminal of MEL moves through the hydrophobic tail region (see Figure 9d,e) and experiences a permeation barrier at the centre of the bilayer. It is due to the fact that the residues of MEL near the C-terminal (i.e., LYS_21_, ARG_22_, LYS_23_ and ARG_24_) with a net charge of +4 prefer to stay at the upper head group-water interface, whereas the N-terminal residue LYS_7_ with a net charge of +1 prefers to move into the inner head group-tail interface, which eventually leads to an increase of the free energy barrier at the centre of the PLB [62]. When (the N-terminal of) MEL traverses further towards the lower leaflet, the free energy drops again, because most of the hydrophobic residues of MEL stay in the hydrophobic region of the PLB. Finally, at the lower head group region of the PLB, the energy rises again, because the charged residue LYS_7_ near the N-terminal of MEL binds with the head group of the lower leaflet, resulting in a new barrier against translocation [40]. Moreover, as MEL penetrates through the bilayer, its conformation changes, but most of its helical structure is retained at the surface of the PLB (see Figure 9b–d).

It is important to notice that the first free energy maximum for the native case (see Figure 9a) is 29.28 ± 1.04 kJ/mol, obtained at the centre of the bilayer (i.e., *z* = 0 nm) and the second maximum is 24.9 ± 2.3 kJ/mol, obtained at around *z* = −2 nm, which indicates that MEL has to face multiple barriers across the bilayer, making the translocation more difficult. In addition, we observed an asymmetric free energy profile, that is most likely due to the disturbance and conformational changes in the PLB, which was observed in other simulation studies as well [62]. 

On the other hand, in the oxidized PLB, the energy reaches a minimum in the centre of the bilayer, because when the lipid tails are oxidized, they become less apolar and hence the hydrophilicity of the membrane core increases dramatically, so that MEL can penetrate more easily. In addition, on top of the general minimum in free energy, there are multiple free energy local maxima and minima, due to interactions of the amino acid side chains of MEL with the lipid tails.

Previous reports suggest that cancer cells are more susceptible than normal cells to oxidative damage and cell death induced by CAP treatments [26,63,64,65,66]. In silico studies of CAP-induced oxidation of phospholipids in cell membranes demonstrated that cell membranes with higher fractions of cholesterol (i.e., normal cells) were protected from pore formation. In contrast, when lower concentrations of cholesterol were present (i.e., cancer cells), the cell membrane was more vulnerable to oxidative stress and it favoured pore formation [48]. The pores generated in the cell membrane facilitated the pass of plasma-generated ROS into the intracellular compartment [67,68], where they could exert further oxidative damage to cells. Our simulation results are in agreement with the literature, as we demonstrate here that RONS present in PT-PBS (e.g., H_2_O_2_, NO_3_^−^ and NO_2_^−^ ions [24,30] oxidize the cell membrane (as can be deduced from the lipid peroxidation experiments, see Figure 6). In addition, from Figure 9a we conclude that the free energy barrier for the transport of MEL across the PLB decreases upon oxidation, and even disappears in large enough oxidation degrees. Hence, the probability of MEL permeation to the cell interior increases, so that lower doses of MEL already give cytotoxic effects. As cancer cells are more sensitive to oxidative stress than normal cells due to the high levels of steady-state RONS produced [69], anticancer treatments (such as the one described here) that increase the oxidative stress levels beyond the RONS threshold could aid to the elimination of cancer cells without damaging the normal cells [70]. Altogether, our results indicate that the effectivity of the combined treatment to eliminate cancer cells relies on the ability of both MEL and PT-PBS to: (i) disrupt the fluidity and integrity of the cell membrane; (ii) facilitate their access into the intracellular compartment; (iii) damage the intracellular components of cancer cells to induce cell death. Due to the differences in the lipid composition of cell membranes [48] and steady-state ROS levels produced by normal and cancer cells, normal cells are expected to be less susceptible to the lipid peroxidation and pore formation induced by the combined treatment. An insufficient damage to the cell membrane integrity of normal cells would mean a reduction in the access of PT-PBS and MEL into the cell, thus limiting their cytotoxic effects. Further studies are required to ensure the safety of the combination of MEL and PT-PBS for the future treatment of cancer patients. 

## 4. Conclusions

Bee venom is considered as a potential weapon against cancer. MEL, a major polypeptide of bee venom, is thought to function as lytic agent, and has been used traditionally in various cancer therapies [2]. However, at high treatment doses, it exhibits severe nonspecific toxicity, because of its lytic properties [3]. In this paper, we describe for the first time a combined treatment of MEL with plasma-treated PBS that reduces the effective dose of MEL required to eliminate cancer cells. We have used in vitro, in ovo and in silico approaches to investigate the synergistic effect of PT-PBS and MEL on A375 melanoma and MCF7 breast cancer cells. We have previously demonstrated that CAP-treated liquids such as PBS, deionized water and culture media, can effectively eliminate pancreatic cancer, melanoma and glioblastoma cancer cells [24,42,71]. In these studies, H_2_O_2_ and NO_2_^−^ present in CAP-treated liquids play a key role in the induction of cell death in cancer cells. These RONS can further favour the formation of ONOO^−^ [24,42,63], which has a high potential to induce lipid peroxidation [72]. Our findings suggest that the ability of the RONS present in PT-PBS to damage the lipids in the cell membrane increases the MEL permeation and therefore lowers the therapeutic dose of MEL required to exert a cytotoxic effect in cancer cells. Our study demonstrates the synergistic effect of MEL and PT-PBS, i.e., the combined cancer cell cytotoxicity for both A375, MCF7 cell lines and malignant melanoma cancer tumours is larger than the sum of both individual effects of MEL and PT-PBS at the same concentration. This is supported by the calculated combination index (CI) values of 0.4, which indicates synergism. This synergy is attributed to the change in membrane integrity upon peroxidation of the membrane lipids by the RONS present in PT-PBS. As a result of the change in membrane integrity, the MEL translocation rate through the cell membrane increases significantly, as demonstrated by US MD simulations. Hence, MEL will already be able to penetrate inside the cell interior at lower treatment doses. Our results show that this synergy between PT-PBS and MEL has great potential for cancer therapy based on MEL, as it might help to reduce the non-specific toxicity of MEL.

## Figures and Tables

**Figure 1 cancers-11-01109-f001:**
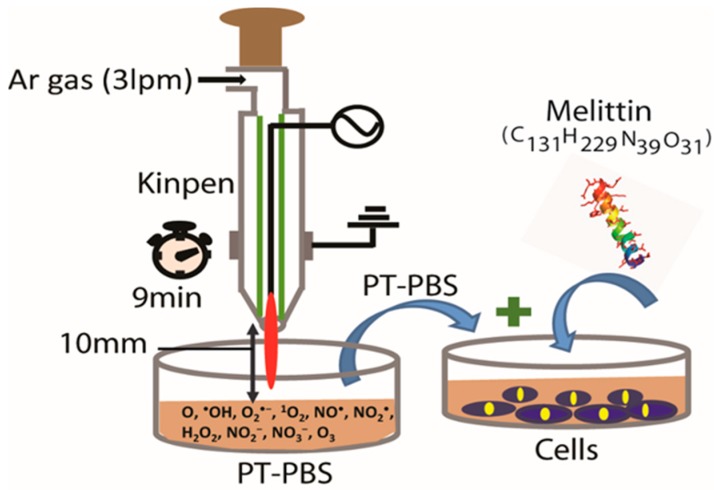
Schematic diagram of the kINPen^®^ IND device. Argon is used as feeding gas with a flow rate of 3 lpm (liter per minute).

**Figure 2 cancers-11-01109-f002:**
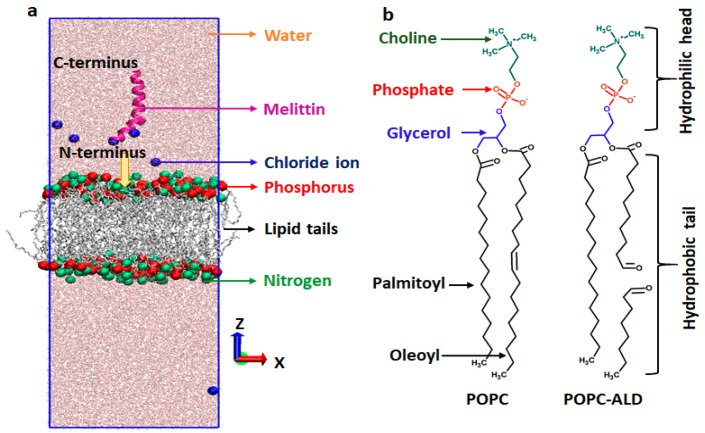
(**a**) Intact (or native) palmitoyl-oleoyl-phosphatidylcholine phospholipid bilayers (POPC PLB), together with melittin (MEL) in the water region. For the sake of clarity, the N and P atoms of POPC are shown with bigger beads and the lipid tails are in grey. The yellow arrow indicates the pulling direction of MEL. (**b**) Schematic illustration of native (POPC) and oxidized (into aldehyde; POPC-ALD) phospholipids. The head group consists of choline, phosphate and glycerol, whereas the lipid tails are two fatty acid chains.

**Figure 3 cancers-11-01109-f003:**
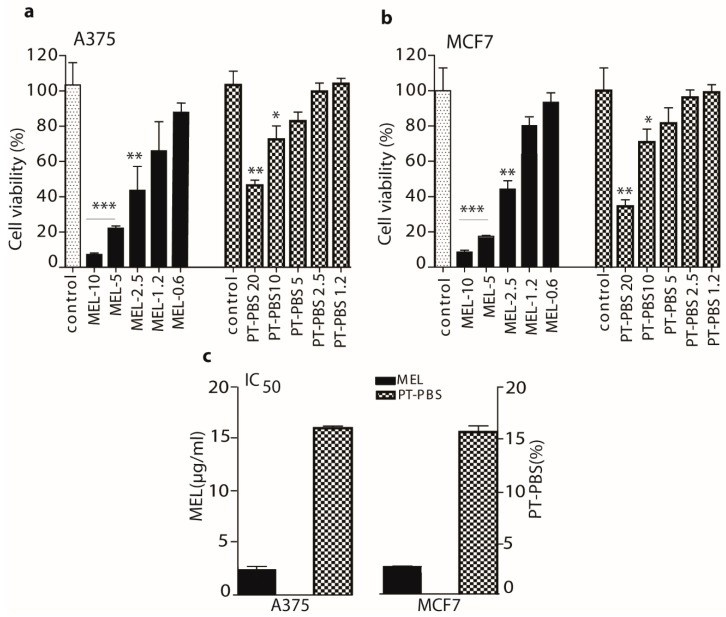
Estimation of optimal doses of MEL and PT-PBS alone, for the cytotoxicity of A375 and MCF7 cells. We measured the cell viability of (**a**) A375 cells, and (**b**) MCF7 cells, at different doses of MEL and PT-PBS, after 24 h incubation. (**c**) Half maximal inhibitory concentration (IC_50_) values of MEL and PT-PBS. Data shown as mean ± SD; * = *p* ≤ 0.05; ** = *p* ≤ 0.01; *** = *p* ≤ 0.001.

**Figure 4 cancers-11-01109-f004:**
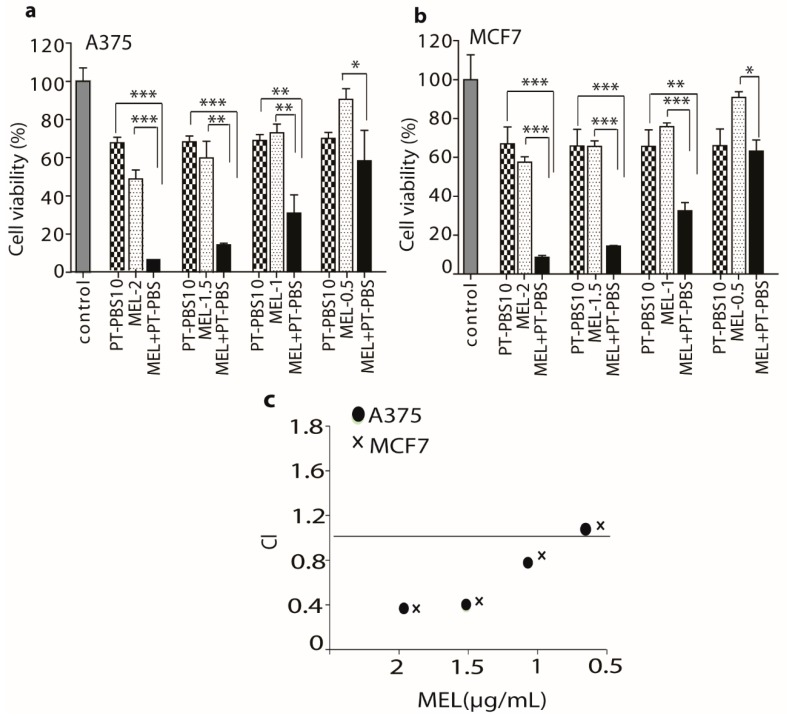
Analysis of cell viability of (**a**) A375 and (**b**) MCF7 cells treated with a fixed dose of PT-PBS (10%), and varying doses of MEL (i.e., 2, 1.5, 1 and 0.5 µg/mL) alone and in combination, 24 h after treatment. (**c**) Combination index (CI) of PT-PBS (10%) with MEL (at 2, 1.5, 1 and 0.5 µg/mL), in A375 and MCF7 cells (see text). All values are expressed as mean ± SD; * = *p* ≤ 0.05; ** = *p* ≤ 0.01; *** = *p* ≤ 0.001.

**Figure 5 cancers-11-01109-f005:**
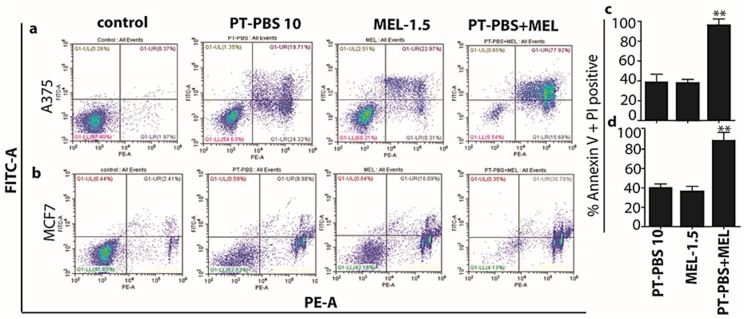
Flow cytometry analysis of the (**a**) A375 and (**b**) MCF7 untreated control cells, or treated with PT-PBS 10, MEL-1.5, and combined treatment (PT-PBS 10 and MEL-1.5). Percentage of cell death in (**c**) A375 and (**d**) MCF7 upon treatment with PT-PBS 10, MEL-1.5 and in combination. All values are expressed as mean ± SD; ** = *p* ≤ 0.01.

**Figure 6 cancers-11-01109-f006:**
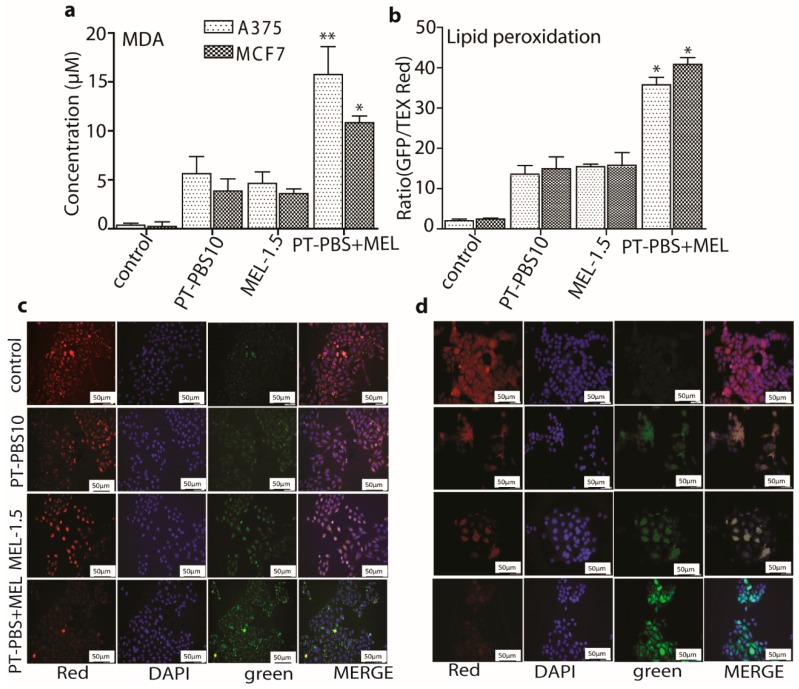
Change in membrane integrity upon PT-PBS 10, MEL-1.5 and combined treatment in both cancer cell lines, by (**a**) measurement of the concentration of the peroxidation product malondialdehyde (MDA), and (**b**) flow cytometry analysis with a fluorescent probe. All values are expressed as ± SD; * = *p* ≤ 0.05; ** = *p* ≤ 0.01. Fluorescence images of (**c**) A375 and (**d**) MCF7 cells, stained with lipid/lipid peroxide detection reagents. Red fluorescence represents non-oxidized membrane lipids, DAPI (blue) represents the nuclear counterstain and green represents the oxidized membrane lipids. Scale bars = 50 µm.

**Figure 7 cancers-11-01109-f007:**
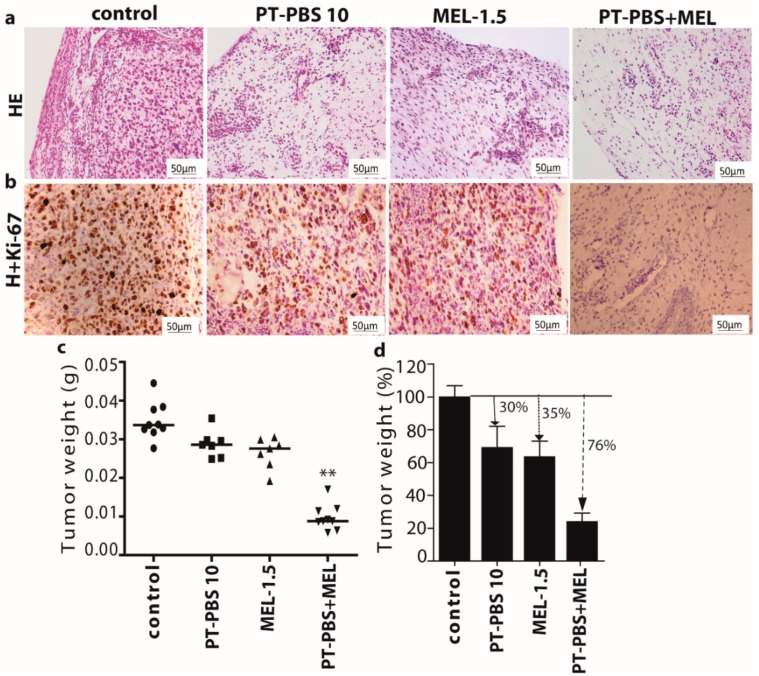
The combined treatment reduced the tumour size and expression of the proliferative marker Ki-67 in melanoma cancer tumours in ovo. (**a**) Representative images of HE and (**b**) Ki-67 staining of tumours exposed to PT-PBS 10, MEL-1.5 or combined treatment. Scale bars represent 50 µm. (**c**) Weight reduction upon treatments in tumours in ovo. Each dot represents one tumour. (**d**) Quantification of tumour weight in percentage (treated/control * 100%) after treatment (PT-PBS 10, MEL-1.5 or combined treatment). Vertical arrows indicate reduction in tumour weight. All values are expressed as ± SD; ** = *p* ≤ 0.01.

**Figure 8 cancers-11-01109-f008:**
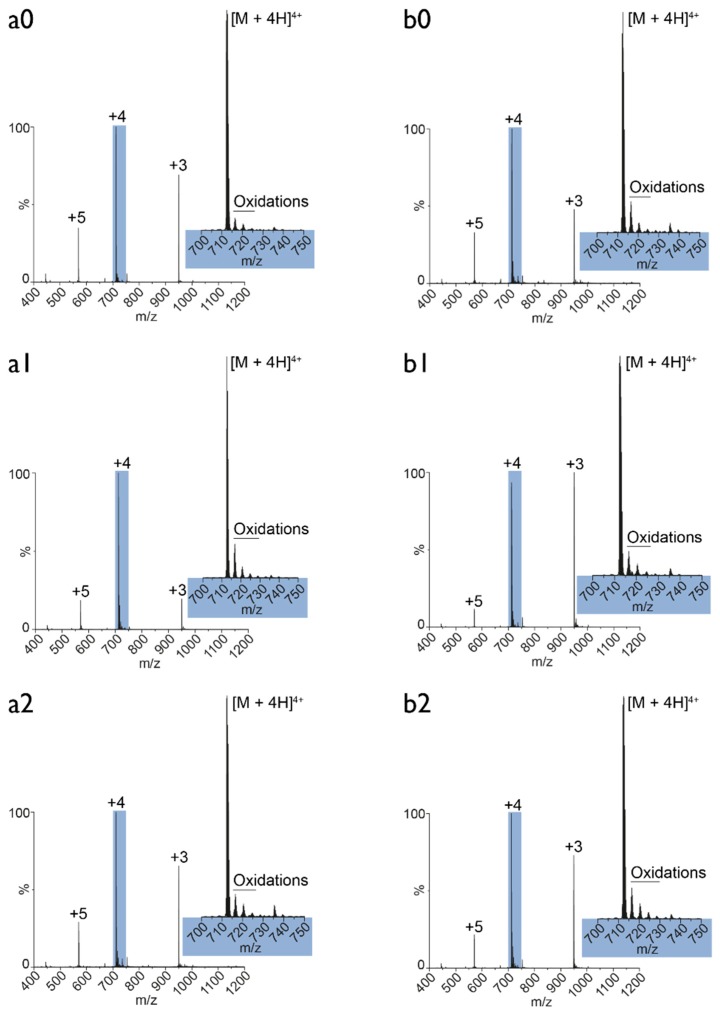
Native mass spectrometry (MS) of MEL. The figure shows an overview of native MS spectra of MEL solubilized either in (**a**) untreated or (**b**) plasma-treated PBS buffer after different time points of incubation (0: 1 min; 1: 60 min; 2: 120 min). The inset in each spectrum displays the *m*/*z* region from 700 to 750 (zoom-in of the [M + 4H]^4+^ species). The satellite peaks highlighted in the insets correspond to oxidations of MEL.

**Figure 9 cancers-11-01109-f009:**
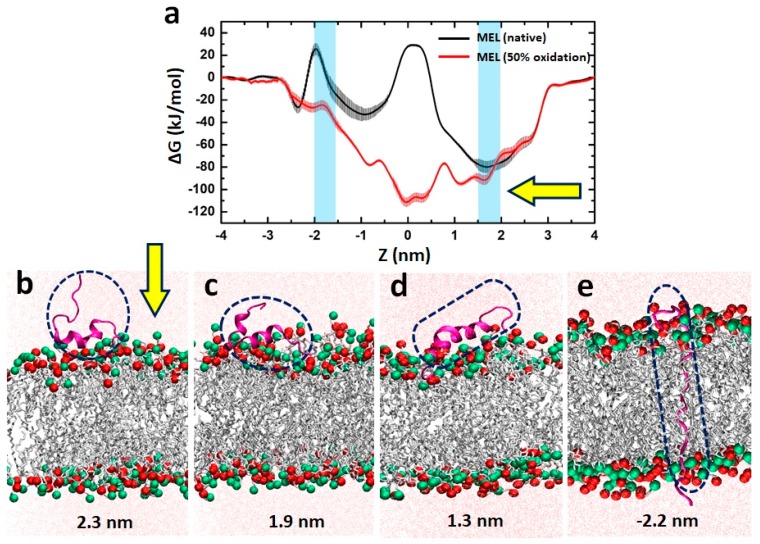
(**a**) Free energy profiles for translocation of MEL across the native and 50% oxidized PLB. (**b**–**e**) MEL at different positions of the native PLB. P and N atoms in the PLB are shown in red and green, respectively. MEL (shown within dashed circles/ovals) and the lipid tails are presented in magenta and grey, respectively. The light blue colour in (**a**) represents the upper (right) and lower (left) head groups of the PLB. The yellow arrows in (**a**,**b**) indicate the direction of MEL translocation. At the bottom of (**b**–**e**), the respective positions of the N-terminal of MEL are indicated, referring to the x-axis of (**a**).

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
