# Peer review of "Synergistic Effects of Melittin and Plasma Treatment: A Promising Approach for Cancer Therapy"

_cancers, 2019, doi:10.3390/cancers11081109_

Round 1
Reviewer 1 Report
Comments and Suggestions for Authors
The article entitled “Synergistic effects of Melittin and plasma treatment: A promising approach for cancer therapy” addresses an interesting subject about the effect of melittin against tumor-lines in an oxidized environment from an integrative perspective (both experimental and computational work). The work is well presented: abstract, introduction, discussion, methods and conclusions are coherent. The methodology employed is adequate for the aims and the results obtained seems to be promising and authors even discussed the putative mechanisms of action involved in the reduction of the side effects produced by melittin in the studied conditions. The conclusions are well supported by the methods, results and the putative mechanisms proposed by the authors.
Minor comments
Line 45 authors said that “MEL with chemotherapeutic drugs and nanotechnology have been previously reported, but this combination is still a challenge”. In this sense, authors cited [19]. Bei, C.; Bindu, T.; Remant, K.C.; Peisheng, X. Dual secured nano-melittin for the safe and effective eradication of cancer cells. Journal of materials chemistry. B 2015, which recognized the notorious side effect produced by amphipathic peptides (e.g., melittin) in red blood cells. In addition, in the previous work cited, they developed an environment-sensitive peptide delivery system (dual secured nano-sting (DSNS), that killed almost 100% of MCF-7, HCT-116, SKOV-3, and NCI/ADR-RES (multidrug resistant) cancer cells at the concentration of 5 μM, without any hemolytic effect. However, authors don’t explain the previous sentences to clarify the state of the art the aforementioned treatment (MEL with chemotherapeutic drugs) which can lead readers to understand what novelty in this study should be. On the other hand, at the end of the introduction (line 80), authors cited works that previously study related to MD simulations approaching the interaction of MEL with PLB, but clearly stated that the free energy barrier of MEL in an oxidized PLB was not yet investigated. For this reason, I encourage the authors to make minor changes in the Introduction
Line 103 : there is not clear the unit of the flow rate used
Line 120: see the symbol used for temperature
Author Response
Please find a point-by-point response to the reviewers’ comments below in blue:
Reviewer #1:
Line 45 authors said that “MEL with chemotherapeutic drugs and nanotechnology have been previously reported, but this combination is still a challenge”. In this sense, authors cited [19]. Bei, C.; Bindu, T.; Remant, K.C.; Peisheng, X. Dual secured nano-melittin for the safe and effective eradication of cancer cells. Journal of materials chemistry. B 2015, which recognized the notorious side effect produced by amphipathic peptides (e.g., melittin) in red blood cells. In addition, in the previous work cited, they developed an environment-sensitive peptide delivery system (dual secured nano-sting (DSNS), that killed almost 100% of MCF-7, HCT-116, SKOV-3, and NCI/ADR-RES (multidrug resistant) cancer cells at the concentration of 5 μM, without any hemolytic effect. However, authors don’t explain the previous sentences to clarify the state of the art of the aforementioned treatment (MEL with chemotherapeutic drugs) which can lead readers to understand what novelty in this study should be. On the other hand, at the end of the introduction (line 80), authors cited works that previously study related to MD simulations approaching the interaction of MEL with PLB, but clearly stated that the free energy barrier of MEL in an oxidized PLB was not yet investigated. For this reason, I encourage the authors to make minor changes in the Introduction
Response 1: Response for Point 1. (In Blue)
We thank the reviewer for finding our manuscript interesting. We have clarified this point in the text, see page 2, line 50 to 60, in the Introduction section, about combined treatment of MEL with chemotherapeutic drugs, which is helpful for the readers to understand the novelty in this study. To take the reviewer’s comment into account, we explained the effects of MEL in combination with chemotherapeutic drugs based on literature, so we now included the following sentences in the text:
“Additionally, Orsolic and Alonezi et al., explored the dose dependent growth-inhibiting impact of MEL in conjunction with a cytotoxic drugs such as cisplatin and bleomycin on melanoma, HeLa and V79 cells in vitro [20,21]. Further, Alizadehnohi et al. reported that MEL enhanced the cytotoxic impact of cisplatin in human ovarian cancer cells [22]. However, the combined use of MEL and cisplatin to treat cancer cells still remained a challenge due to the side effects and off-target toxicity [19,23]. It has been suggested that the combination of MEL with nanoparticles could increase the safe delivery of significant amounts of MEL intravenously to target and kill tumors, while reducing the hemolytic activity of MEL [1,15]. However, the role of nanotechnology in delivering MEL is still at its early development stage because of drawbacks during the preparation for nano delivery systems such as aggregation, morphological changes, peptide stability, etc. Moreover, these system are expensive to implement for cancer therapy [24].].”.
20. Oršolić, N. Potentiation of Bleomycin lethality in HeLa and V79 cells by bee venom. Archives of Industrial Hygiene and Toxicology 2009, 60, 317-326.
21. Alonezi, S.; Tusiimire, J.; Wallace, J.; Dufton, M.; Parkinson, J.; Young, L.; Clements, C.; Park, J.-K.; Jeon, J.-W.; Ferro, V. Metabolomic profiling of the synergistic effects of melittin in combination with cisplatin on ovarian cancer cells. Metabolites 2017, 7, 14.
22. Alizadehnohi, M.; Nabiuni, M.; Nazari, Z.; Safaeinejad, Z.; Irian, S. The synergistic cytotoxic effect of cisplatin and honey bee venom on human ovarian cancer cell line A2780cp. Journal of venom research 2012, 3, 22.
23. Rady, I.; Siddiqui, I.A.; Rady, M.; Mukhtar, H. Melittin, a major peptide component of bee venom, and its conjugates in cancer therapy. Cancer letters 2017, 402, 16-31.
24. Biswaro, L.S.; da Costa Sousa, M.G.; Rezende, T.; Dias, S.C.; Franco, O.L. Antimicrobial peptides and nanotechnology, recent advances and challenges. Frontiers in microbiology 2018, 9, 855.
In addition, we added a sentence at the end of the introduction, to clarify better that the papers in literature on MD studies only applied to native PLBs, and not to oxidized PLBs, as produced by plasma oxidation. See page 2 lines 95 to 96
Line 103 : there is not clear the unit of the flow rate used
Response 2: Response for Point 2. (In Blue)
We have corrected the flow rate of gas on page 3 line 121 (lpm = liter per minute).
Line 120: see the symbol used for temperature
Response 3: Response for Point 3. (In Blue)
Corrected on page 4, line 140
Reviewer 2 Report
It is well known from the current literature that melittin is suggeste as a promising anti-cancer molecule (e.g. Gajski G, Garaj-Vrhovac V. Melittin: a lytic peptide with anticancer properties. Environ Toxicol Pharmacol. 2013 Sep;36(2):697-705; Liu CC, Hao DJ, Zhang Q, An J, Zhao JJ, Chen B, Zhang LL, Yang H. Application of bee venom and its main constituent melittin for cancer treatment. Cancer Chemother Pharmacol. 2016 Dec;78(6):1113-1130; Rady I, Siddiqui IA, Rady M, Mukhtar H. Melittin, a major peptide component of bee venom, and its conjugates in cancer therapy. Cancer Lett. 2017 Aug 28;402:16-31, and so forth) therefore the paper does not represent a true novelty in th field. What is very strange is the use of plasma-treated phosphate buffered saline solutions (PT-PBS), the composition of which and the related activity was not deepen.
Furthermore, statistics is poorly described and applied.
Author Response
Please find a point-by-point response to the reviewers’ comments below in blue:
Reviewer #2:
It is well known from the current literature that melittin is suggested as a promising anti-cancer molecule (e.g. Gajski G, Garaj-Vrhovac V. Melittin: a lytic peptide with anticancer properties. Environ Toxicol Pharmacol. 2013 Sep;36(2):697-705; Liu CC, Hao DJ, Zhang Q, An J, Zhao JJ, Chen B, Zhang LL, Yang H. Application of bee venom and its main constituent melittin for cancer treatment. Cancer Chemother Pharmacol. 2016 Dec;78(6):1113-1130; Rady I, Siddiqui IA, Rady M, Mukhtar H. Melittin, a major peptide component of bee venom, and its conjugates in cancer therapy. Cancer Lett. 2017 Aug 28;402:16-31, and so forth) therefore the paper does not represent a true novelty in the field. What is very strange is the use of plasma-treated phosphate buffered saline solutions (PT-PBS), the composition of which and the related activity was not deepen.
Response 1: Response for Point 1. (In Blue)
We agree with the reviewer that MEL has been suggested as a promising anti-cancer molecule in literature. However, it is also reported that high doses of MEL cause several issues, including non-specific cytotoxicity, degradation and off-target toxicity (see references 10 - 13 ). Due to these side effects, Gajski et al., Liu et al., and Rady et al., (i.e., the papers mentioned by the reviewer) explored the delivery of MEL with nanoparticles to improve the antitumor efficiency and therapeutic capabilities of MEL. This approach is however still challenging, because the ability of nanoparticles to deliver MEL is still in an early stage of study. Additionally, the therapy is expensive for patients (as mentioned in the introduction section, at page 2, line number 50 to 60). The novelty of our study relies on the use of an inexpensive alternative treatment, i.e., based on cold atmospheric plasma (CAP). CAP is an ionized gas, which consists of a cocktail of reactive species (e.g., reactive oxygen and nitrogen species: RONS), and it has been demonstrated in multiple studies to be effective against cancer. In addition, it can be used to treat solutions (e.g., plasma-treated phosphate-buffered saline: PT-PBS), which get similar anti-cancer effects, and when used in combination with MEL, it might subjugate the non-specific cytotoxicity problems of MEL. Our study aims to investigate whether the combined treatment of PT-PBS and MEL can have synergistic effect on A375 melanoma and MCF7 breast cancer cells, as well as in an egg tumor model, so that the effective therapeutic dose of MEL can be lowered, thereby reducing its side effects. Such study has never been performed before, and is thus truly novel.
To account for the reviewer’s comment, we have now explained in more detail that the RONS present in PT-PBS are responsible for the biological effects observed in cancer cells
See page 4, line number 132 to 134:
“The RONS present in plasma-treated PBS have been previously described [20], being H2O2 and NO2- as two of the main RONS responsible of the biological effects of PT-PBS.”
And see conclusion section, page 16, line number 506 to 513:
“We have previously demonstrated that CAP-treated liquids, such as PBS, deionized water and culture media, can effectively eliminate, pancreatic cancer, melanoma and glioblastoma cancer cells [25,43,64] In these studies, H2O2 and NO2- present in CAP-treated liquids play a key role in the induction of cell death in cancer cells. These ROS can further favour the formation of ONOO− [25,43,64], which has a high potential to induce lipid peroxidation [65] . Our findings suggests that the ability of the RONS present in PT-PBS to damage the lipids in the cell membrane increases the MEL permeation and therefore lowers the therapeutic dose of MEL required to exert a cytotoxic effect in cancer cells.”.
64. Privat-Maldonado, A.; Gorbanev, Y.; Dewilde, S.; Smits, E.; Bogaerts, A. Reduction of Human Glioblastoma Spheroids Using Cold Atmospheric Plasma: The Combined Effect of Short- and Long-Lived Reactive Species. Cancers 2018, 10, 394.
65. Radi, R.; Beckman, J.S.; Bush, K.M.; Freeman, B.A. Peroxynitrite-induced membrane lipid peroxidation: the cytotoxic potential of superoxide and nitric oxide. Archives of biochemistry and biophysics 1991, 288, 481-487.
Furthermore, statistics is poorly described and applied.
Response 2: Response for Point 2. (In Blue)
According to the reviewer’s comment, we have explicitly added the statistical analysis, as mentioned in page 6, line 228 to 231:
“Data was analysed using the Students’-test comparison analysis. The data was considered significantly different when * = p ≤ 0.05, ** = p ≤ 0.01, *** = p ≤ 0.001. All values represent experiments done in triplicates. Data shown as mean ± standard deviation (SD). Prism (Graphpad Software Inc.) and Excel Software (Microsoft Inc.) was used to compare the groups.”
We have also added in the captions of each figure that data has been processed for statistical analysis (mentioning mean ± SD values, p values).
We also improved the English writing in the introduction, methodology, results, and conclusion section, throughout the manuscript, and the corresponding improvements are highlighted in yellow in the manuscript.